# ALT Positivity in Human Cancers: Prevalence and Clinical Insights

**DOI:** 10.3390/cancers13102384

**Published:** 2021-05-14

**Authors:** Danny MacKenzie, Andrea K. Watters, Julie T. To, Melody W. Young, Jonathan Muratori, Marni H. Wilkoff, Rita G. Abraham, Maria M. Plummer, Dong Zhang

**Affiliations:** Department of Biomedical Sciences, College of Osteopathic Medicine, New York Institute of Technology, Old Westbury, NY 11568, USA; dmackenz@nyit.edu (D.M.J.); awatters@nyit.edu (A.K.W.); jto01@nyit.edu (J.T.T.); myoung08@nyit.edu (M.W.Y.); jmurator@nyit.edu (J.M.); mwilkoff@nyit.edu (M.H.W.); rabrah08@nyit.edu (R.G.A.)

**Keywords:** alternative lengthening of telomeres, cancers, ALT biomarkers, ATRX, DAXX

## Abstract

**Simple Summary:**

Since it was first described over two decades ago, the Alternative Lengthening of Telomeres (ALT) pathway has been well accepted to hold clinical significance in cancer development, cancer diagnosis, and cancer treatment. In this review, first, we discuss how the activation of this pathway is determined. Then, we provide up-to-date statistics on the cancers ALT activity is detected in. We discuss the relationships between ALT positivity and prognosis as well as the pathogenetics of ALT positive cancers. Finally, we evaluate pre-clinical and clinical investigations of potential therapies targeting ALT.

**Abstract:**

Many exciting advances in cancer-related telomere biology have been made in the past decade. Of these recent advances, great progress has also been made with respect to the Alternative Lengthening of Telomeres (ALT) pathway. Along with a better understanding of the molecular mechanism of this unique telomere maintenance pathway, many studies have also evaluated ALT activity in various cancer subtypes. We first briefly review and assess a variety of commonly used ALT biomarkers. Then, we provide both an update on ALT-positive (ALT+) tumor prevalence as well as a systematic clinical assessment of the presently studied ALT+ malignancies. Additionally, we discuss the pathogenetic alterations in ALT+ cancers, for example, the mutation status of *ATRX* and *DAXX*, and their correlations with the activation of the ALT pathway. Finally, we highlight important ALT+ clinical associations within each cancer subtype and subdivisions within, as well as their prognoses. We hope this alternative perspective will allow scientists, clinicians, and drug developers to have greater insight into the ALT cancers so that together, we may develop more efficacious treatments and improved management strategies to meet the urgent needs of cancer patients.

## 1. Introduction

### 1.1. Telomere and Telomere Maintenance Mechanisms

Human telomeres consist of repetitive DNA sequences of (TTAGGG)_n_ at the terminal ends of each linear chromosome. The normal length of human telomeres ranges between 10 kilobases (kb) and 15 kb. Despite the high fidelity of DNA replication machinery, DNA ends progressively shorten with each cell division in a process termed the “End Replication Problem” [1]. Though this loss occurs in most somatic cells, telomeres protect coding DNA from attrition through structural barriers. Condensed primarily as heterochromatin, telomeres are also associated with a variety of proteins, including a six-member protein complex called Shelterin. The primary functions of Shelterin include protecting telomeres, facilitating telomere synthesis, and modulating the DNA damage response (DDR) at telomeres, among others [2,3]. Typically, at the telomere terminus, the G-rich strand extends past its C-rich complementary strand and forms the so-called “G-overhang”. The 3′ G-overhangs fold back and invade upstream telomeric repeats to form lasso-like secondary structures termed “T-loops” [3,4]. Together with Shelterin, T-loops stabilize the terminal ends of DNA and prevent them from being recognized as double-stranded breaks (DSBs).

Neoplastic cells typically activate one of the two Telomere Maintenance Mechanisms (TMM) to maintain their telomeres during uncontrolled proliferation. Most tumors reactivate telomerase, a high-fidelity DNA transferase with reverse transcriptase activity [5,6]. This enzyme consists of telomerase RNA component (hTR), human telomerase reverse transcriptase (hTERT) protein subunit, as well as other accessory proteins. Telomerase adds the TTAGGG one at a time to telomeres when they are getting too short [7,8]. In 1997, Shay and Bacchetti reported the first large-scale survey of telomerase activity in various tissues using the telomeric repeat amplification protocol (TRAP) assay [5]. This assay is still the gold-standard practice for confirming telomerase activity in tumors. Analysis of their survey demonstrates that: (1) 82.4% of malignant neoplasms (1798/2182) are telomerase-positive (TEL+); (2) 49.1% of premalignant lesions (54/110) are TEL+; (3) 23.6% of benign neoplasia (129/547) are TEL+ [5]. Their data suggested that telomerase upregulation may be closely related to malignant transformation. The current consensus agrees with their estimates, placing telomerase prevalence at 80–90% of all malignancies [2,6,9,10].

A significant percentage of neoplasias activate the second type of TMM, called alternative lengthening of telomeres (ALT), to achieve replicative immortality and telomere elongation. ALT is commonly thought to occur in about 10–20% of all tumors [2,6,9]. Unlike the TEL+ tumors, which rely on the enzymatic activity of a single enzyme [11], ALT relies on many DNA damage response (DDR) proteins, including those involved in the homology-dependent repair (HDR) pathway [12,13]. Since telomeric sequences are redundantly repetitive within all chromosomes, the ALT-positive (ALT+) tumors use these repeats to induce strand invasion of one telomeric strand into another. The newfound complementary sequences then serve as elongation templates for the invading strand. This homology might occur through invasion into the homologous chromosome, or an unrelated chromosome, or even the extrachromosomal telomeric repeats (ECTRs) [4,14]. Shortened telomeres in ALT+ cells can elongate substantially by recruiting specialized replication machinery in a process termed Break Induced Telomere Synthesis (BITS) [2]. Adopting the ALT pathway confers many unique molecular characteristics to tumors, some of which are commonly used as markers of ALT activity. ALT is also associated with certain tumor types with varying impacts on patient prognosis (Figure 1), the details of which will be discussed below.

Most recently, Dagg and colleagues identified a small group of cancers that rely neither on telomerase nor the ALT pathway for their survival [15]. Since these cancers can proliferate in cell culture for more than 200 population doublings while their telomeres keep getting shorter and shorter, they are thus called “Ever-Shorter Telomeres”. How these cancers survive without activating one of the two known TMMs remains a mystery.

### 1.2. Commonly Used ALT Biomarkers

Confirming neoplasms as ALT+, historically, has not been as simple as defining their TEL+ counterparts. Whereas telomerase positivity relies on quantifying the activity of a single enzyme via the well-established TRAP assay, our current understanding of ALT is that there is no apparent singular identifying enzyme or characteristic to rely on. In fact, identification of ALT depends on a tumor displaying multiple of ALT’s many defining characteristics. In addition to the absence of telomerase and/or telomerase activity, ALT features also include, as shown in Table 1, the presence of: (1) telomeres of substantial length (>50 kb) [16]; (2) telomeres of heterogeneous length (<8 kb and >50 kb) [16,17]; (3) elevated levels of telomere sister chromatid exchange (tSCE) [18]; (4) extra-chromosomal telomeric repeats, or ECTRs, particularly C-circles [19,20]; (5) ALT-associated acute promyelocytic leukemia (PML) bodies, or APBs [21]; (6) telomere dysfunction-induced foci, or TIFs [22].

In general, ALT has a heterogeneous phenotype that is often hard to quantitate; so much so that ALT has been proposed to have multiple pathways [23]. Thus, the best identification of an ALT+ cancer involves confirmation via multiple biomarkers. However, this ideal standard is rarely met, especially in large-scale ALT prevalence studies, likely due to limited tissue availability, time, technical expertise, or other resources. To best review the recent surge in ALT prevalence literature, careful evaluation of how ALT is identified becomes very important.

Telomere length and heterogeneity were among the first biomarkers used to identify ALT+ samples [16,17]; however, they have seen limited use in tumor TMM screening. Unlike most TEL+ cells, which often show a narrow distribution in telomere length, ALT+ cells manifest a much wider range in telomere lengths and contain super long telomeres that can rise upward of 50 kb [16,24]. Such findings are obtained by the Terminal Restriction Fragment (TRF) analysis, a technique that uses endonuclease-mediated cleavage of non-telomeric DNA to enrich telomeric DNA, which is then detected using a Southern Blot. Though well established and widely accepted, this biomarker is rarely used in large-scale ALT identification studies (Appendix A). TMM heterogeneity along with adjacent normal tissue often obscure fragmentation results, which complicates TRF ALT identification [24]. Further, length heterogeneity does not always correlate with other ALT biomarkers, such as APB presence, because heterogeneity is proposed to occur after APB formation [23]. Finally, the use of the TRF method [24] requires properly cryo-preserved samples—a great limitation to most paraffin-embedded tissue libraries.

Similarly limited in feasibility to tumor TMM screening is the measurement of tSCEs. These telomeric recombination events are measured by the use of chromosome orientation-fluorescent in-situ hybridization (CO-FISH), a specialized hybridization technique that detects the orientation of telomere leading and lagging strand probes on sister chromatids [18,24]. Although tSCEs are generally ALT-specific phenomena, certain TEL+ cell lines have been observed with similar tSCE levels [24]. Additionally, CO-FISH requires collecting mitotic cells, which is an innate barrier for using tSCE-based TMM identification in large patient cohorts. For these reasons, the use of tSCEs is limited to cultured cells and is best used as supporting evidence for other ALT biomarkers; we found no large ALT prevalence studies in the past two decades that used tSCEs as part of their identification criteria (Appendix A).

Elevated telomeric damage and heightened DNA repair activity are central to several methods of ALT identification, including the detection and quantification of TIFs and APBs. A hallmark of ALT+ tumors is spontaneous telomeric DNA damages, likely due to frequently stalled replication forks [25,26,27,28]. If stalled replication forks fail to be restarted on time, they are likely to collapse and lead to the generation of DSBs. ALT likely uses these strand breaks to initiate HDR and telomere elongation. Telomeric DNA damage is commonly identified as punctate nuclear foci when cells are stained with an antibody recognizing either 53BP1 or the phosphorylated histone variant H2AX (γH2AX). When these sites of DNA damage colocalize with the telomere via Telomere FISH (Telo-FISH) or with components of the Shelterin complex, such as TRF1 or TRF2, these sites are thus known as Telomere Dysfunction-Induced Foci, or TIFs [2,22]. Relative to TEL+ cells, ALT+ cells typically demonstrate greater intensity and frequency of TIFs; however, this is not always the case [2]. Though associated with ALT, TIFs are not specific enough to identify ALT on their own as they are only markers of telomeric DNA damage, which can occur due to many reasons. However, TIFs do largely colocalize with APBs in ALT+ cells.

APBs are specialized PML nuclear bodies thought to be the primary sites of ALT activity—the locations where damaged telomeres are clustered and repaired [21,29]. Their identification relies on immunohistochemical colocalization of PML protein with Shelterin subunits or with telomeres by Telo-FISH. APBs are among the pre-eminent ALT identification modalities because the resulting telomeric foci are highly abundant in ALT+ cells and tumors [24]. They colocalize with telomeres, DNA damage markers, as well as with HDR and BITS machinery [30]. Mechanistically, APBs serve as scaffolds for protein recruitment [31], drive telomere and ALT protein condensation in a liquid-liquid phase separation (LLPS) manner via Small Ubiquitin-like Modifier (SUMO) and SUMO-interacting motifs (SIM) [32,33], and are thought to be the sites where telomeric HDR and elongation occur [24]. Despite the close associations with ALT, APB foci are also observed occasionally in TEL+ cells [34] and not ubiquitously observed in all ALT+ cells [35,36,37]. Despite this limitation, APB identification allows convenient TMM classification in large cohorts. Early studies extensively validated the use of Telo-FISH with PML immunofluorescence as a method for identifying tumors as ALT+ [38].

Recently, visualization of APBs has been combined with cell synchronization and fluorescently labeled nucleotides to detect ongoing telomeric DNA synthesis [39]. Termed the ALT Telomere Synthesis in APBs (ATSA) assay, this technique involves trapping enriching cells in G2, which is the cell cycle phase most associated with telomeric repair and synthesis. It is a direct assessment of ALT positivity and activity. However, it is limited in use for tumor screening as it requires synchronizing cells.

Ultra-bright telomeric foci identified by Telo-FISH alone became an increasingly acceptable biomarker for defining ALT positivity when Heaphy and colleagues published the first comprehensive assessment of ALT prevalence across a large set of 94 malignant tumor subtypes [40]. The study estimated ALT’s total prevalence to 4% in a sample size of 6110 malignancies, which is intriguingly lower than the commonly cited 10–20% as mentioned above [40]. The reason for the discrepancy warrants further investigation. The criteria for ALT positivity required identifying ultra-bright telo-FISH nuclear foci within cells, then quantifying positive-cell prevalence within a sample. If the overall prevalence of ultra-bright foci was greater than 1%, tumors were identified as ALT+. Given the established overlap of TIFs and APBs in ALT+ cells, ultra-bright telo-FISH nuclear foci alone may be sufficient to identify ALT positivity in tumors. Not all telomerase-negative (TEL−) cell lines exhibit APBs or other ALT characteristics [35,36,37], which suggests that PML staining likely underestimates ALT prevalence. Thus, the use of Telo-FISH alone may be a more sensitive identifier or screening tool for identifying ALT+ malignancies. This innovative idea, which is implied by Heaphy and colleagues’ methodology, is limited by their Telo-FISH and APB colocalization studies, which were used as the basis for their novel criteria. They did state that most PML proteins colocalized with ultra-bright foci, though such colocalization was heterogeneous. We tend to agree that this methodology has significant merits, such as its simplicity and scalability. Despite this consensus, quantification of ultra-bright telo-FISH signals that colocalize with APBs should be performed to bring greater credibility and confidence in this popular methodology.

The C-circle assay (CCA) is a sensitive, rapid, and quantitative technique for identifying ALT activity that relies on the measurement of C-circles, a subtype of ECTR, found primarily in ALT+ malignancies [41]. C-circles were proposed to serve as self-priming homology templates that enable rolling circle amplification (RCA) of shortened telomeres [24,41]. However, recent evidence suggests that C-circles are likely the byproducts of BITS during the repair and restart of the stalled or collapsed replication forks at damaged telomeres [27,42]. Telomeric repeats neighboring DNA damage sites are thought to form internal loops (I-loops), which are excised to form ECTRs, such as C-circles [43]. C-circles are regarded as the most sensitive, specific, and quantitative biomarker of the ALT activity [41,44]. When initially developing the CCA, Henson and colleagues found that ALT+ cell lines and tumors, respectively, had 750× and 100× more C-circles than their ALT-negative (ALT−) counterparts and controls [41]. C-circles were barely detectable in any TEL+ cell lines tested but were highly positive in the TEL−cell lines—even those that do not exhibit common ALT characteristics—such as APBs or heterogeneous telomere length [35,41]. In addition to being sensitive and specific, one advantage of the CCA is its quantitative nature, which, unlike other ALT biomarkers, allows the ability to measure changes in ALT activity over time. Therefore, it has potential clinical applicability in quantifying the activity of novel ALT therapeutics or in monitoring ALT cancer progression in patients. Although performing the CCA may involve more technical expertise than telo-FISH alone, the CCA still relies on simple techniques, such as RCA, dot blots, or qPCR [44]. Therefore, CCA is the most highly recommended ALT biomarker for cell lines and tumor biopsies.

### 1.3. Other Potential ALT Identification Strategies

A clear consensus on the preferred or the ideal ALT identification strategy is still lacking. Having reviewed previously published ALT prevalence data on numerous tumors, we have found that the most frequently used biomarkers for ALT identification over the past two decades have been APBs, Telo-FISH, and C-circles (Appendix A). Despite strong evidence suggesting that C-circles are the most sensitive biomarker for ALT [41], Telo-FISH alone has become the most frequently published method of ALT identification in large tumor sample studies. Accepting results from studies using Telo-FISH alone assumes that Telo-FISH correlates with APBs or C-circle findings. An assumption, though not quantified, is generally agreeable when comparing published data (Appendix A).

Whether choosing Telo-FISH, CCA, or APBs, very few studies use more than one ALT biomarker to confirm ALT presence within their samples (Appendix A). Given ALT’s heterogeneous phenotype, the most reliable TMM identification strategy likely involves using multiple biomarkers, including CCA, APBs (or Telo-FISH), and TIFs.

Looking forward, Whole Genome Sequencing (WGS) is currently under consideration as a novel tool for TMM identification. In a 2018 study, Lee and colleagues were the first to validate such an approach [45]. Using machine learning on WGS tumors, they found that the prevalence of telomeric variant repeats, particularly TTTGGG, TAAGGG, and TTAGAG, reliably predicted tumor TMM. They created an ALT classifier that used a random-forest machine learning model and trained it to differentiate tumor TMM using sequence data from their group’s CCA-verified PanNET and melanoma samples. They found the classifier to be 91.6% accurate [45]. Then, after applying the tool to a large panel of 21 tumor subtypes (*n* = 821 samples) from an independent and external WGS center, they confirmed that variant repeat content was sufficient to predict ALT positivity or ALT negativity. ALT prevalence resulted in 21 tumor subtypes, which were consistent with well-established statistics on ALT prevalence [45]. Despite being trained on only PanNETs and melanoma samples, their classifier could still predict ALT positivity in a diverse tumor subtype cohort. Therefore, ALT+ tumors, broadly, are likely to share a common variant repeat sequence profile. Although only tested on 821 samples from a single WGS sequence center, their method is an innovative and promising ALT+ tumor detection method. Further training and validation on larger datasets are warranted.

## 2. ALT Positive Bone Cancers: Osteosarcoma

Osteosarcoma (OS) is the most common primary bone cancer in children and young adults [46]. It is most prevalent in adolescence but can also occur in adulthood with age as a poor prognostic factor [47]. It is considered to be a relatively rare tumor with a reported worldwide incidence of 3.4 cases per million per year [48]. Histologically, OS is a high-grade malignancy of pleomorphic cells that produce unmineralized osteoid or mineralized bone [46,48]. Low-grade OS variants seem to account for only about 1–2% of cases [48]. For pediatric patients, the 5-year overall survival rests at about 68% [49] but has remained unimproved since the advent of modern chemotherapy [50]. Despite this, recent research has uncovered strong OS and ALT associations, which hold potential prognostic and therapeutic value.

OS has long been the standard model by which the ALT pathway is studied. In 2001, Scheel and colleagues surveyed 29 OS malignancies to report only 52% were TEL+ via the TRAP assay, thus implying that about half of OS may use the ALT [51]. They also assessed a panel of seven OS cell lines with TRF assay, including U2OS and Saos-2, which are two of the most commonly used cell lines for investigating the ALT pathway. Follow-up studies used the TRF assay to place OS ALT prevalence at about 68% [38,52,53,54,55]. Our up-to-date survey of published data on OS TMM status estimates ALT prevalence to be around 63% (*n* = 188) (Table 2); however, it is notable that there have only been two contemporary OS ALT prevalence studies since 2005 [55,56]. Neither performed the CCA, and both had a relatively small sample size. Despite rich TMM research using the OS cell lines as the model system, there is a severe paucity in contemporary quantification of ALT prevalence. Updated large-scale analysis of ultra-bright telomeric foci, APBs, and C-circles is urgently needed to further confirm that OS is one of the most prevalent ALT cancers.

TEL-positivity alone confers worse overall survival and progression-free survival (PFS) than the ALT+ OS [54] (Figure 1). In vitro analysis showed that the ALT+ Saos-2 cells are less proliferative and less invasive than the TEL+ OS cells, MSSG-HOS [57]. In an intriguing study, Sanders and colleagues combined the use of two TMM biomarkers, telomere heterogeneity, and hTERT mRNA levels, to describe the ALT+/TEL− tumor profiles, rather than simply TEL− ones [54]. The discriminative analysis found the 3-year PFS of ALT+/TEL− and ALT+/TEL+ to be 62.3% and 50%, respectively. ALT−/TEL+ patients all experienced relapse before 3 years. Another study reported that in OS lacking both TEL and ALT (or TEL−/ALT−), in about 15% of the cases, the 5-year overall survival rate increased to 90%, relative to 60% if one or both TMMs were present [52]. Taken together, current data suggest that the presence of TEL in OS confers the least favorable prognosis. ALT is more favorable and may be protective in the presence of TEL, while the absence of both TEL and ALT confers the most favorable OS prognosis.

### ALT Positive Osteosarcoma and the Mutation Status of ATRX and DAXX

Mutations in the α-thalassemia/mental retardation syndrome X-linked (*ATRX*) and death-domain associated protein (*DAXX*) genes have become intimately linked to the ALT phenotype. Recent OS research suggests an interesting therapeutic implication of re-expression of ATRX and DAXX or reactivating the ATRX-DAXX pathway.

ATRX interacts with DAXX, and together they function to remodel chromatin and deposit certain histone variants, like H3.3 and macroH2A, onto heterochromatin [58,59]. Although their deficiency alone is not sufficient to activate the ALT pathway, mutations in *ATRX* and/or *DAXX* genes are thought to loosen up the heterochromatin structures at the telomeres and facilitate the eventual induction of ALT [60,61,62,63]. The connections between ATRX/DAXX loss and ALT activity were first noted in the genomic studies of PanNETs and CNS tumors [64,65,66]. Those results were further verified when a comprehensive assessment of 22 known ALT+ cell lines found 19 cell lines to be ATRX mutant, including U2OS and Saos-2 [67]. The overall prevalence of ATRX loss in OS, irrespective of TMM, is about 24% (Table 3), which is consistent with ATRX loss in sarcomas overall [68]. Interestingly, ALT+ OS experiences ATRX loss at a frequency of 58% [56], but if ATRX is lost, the OS is ALT+ nearly 100% of the time (Table 3). These estimates are also generally consistent in all sarcomas [68,69]. OS-specific findings are limited by the nature of small sample sizes, but altogether these findings suggest that loss of ATRX is highly specific for ALT.

The prevalence of ATRX/DAXX loss in ALT suggests that re-expressing the two proteins or reactivating the pathway that they are involved in may have therapeutic benefits for ALT+ OS. In a well-controlled study, Clynes and colleagues re-expressed wild-type (WT) ATRX in U2OS and Saos-2 cells and demonstrated the suppression of many ALT phenotypes, including a decrease in C-circles, APBs, telomere length, and tSCEs [60]. However, changes in cell viability were not observed. Most recently, two groups independently characterized a DAXX loss-of-function translocation event in the ALT+ OS cell line, G292 [70,71]. Both studies confirmed that exogenously expressed wild-type DAXX was sufficient to restore the DAXX function and suppress the ALT phenotypes, including the elimination of C-circles and a 4-fold decrease in APBs. Most intriguingly, RNA-Seq analysis showed that expressing WT DAXX in G292 also upregulated the pathway related to osteoblast differentiation [71], suggesting that re-expression of WT ATRX/DAXX in OS cells may stop tumor growth by inducing re-differentiation. Continued validation of this intriguing hypothesis in in vitro and in vivo ALT models are necessary. Nevertheless, inducing tumor differentiation in ATRX- or DAXX-null ALT+ OS is a fascinating idea with potential therapeutic implications worth further inquiry.

## 3. ALT Positive Breast Cancers

Breast cancer is the most frequently diagnosed and second highest cause of cancer-related death in women [49]. It is a heterogeneous cancer with many genetic and immunopathological subtypes. Clinically, breast cancer is classified into four distinct groups: Luminal A, Luminal B, human epidermal growth factor receptor 2 positive (HER2+), and triple-negative breast cancer (TNBC) [72]. Luminal A and Luminal B breast cancers are often estrogen receptor (ER) positive and/or progesterone receptor (PR) positive.

The prevalence of ALT activity in breast carcinomas has been reported to range from 2% to 11% (Appendix A). Updated literature survey narrows this range to between 2% and 8%, with an average of about 4% exhibiting ALT positivity (*n* = 552) (Table 2). Interestingly, two studies have shown that ALT positivity appears to correlate with HER2 overexpression [73,74]. The ALT+/HER2+ breast cancers were first observed to have worse prognoses than the ALT−/HER+ breast cancers by Subhawong and colleagues in 2009 (Figure 1) [73]. Among the 71 breast cancers examined, 21 were HER2+, and of those 21, 3 were ALT+ as indicated by the presence of APBs (14%) [73]. Xu and colleagues reported similar findings in 2013: 6 out of 29 (21%) HER2+ tumors were ALT+ using APBs as the biomarker, while all other molecular subtypes were ALT− [74]. Though only observational, the three ALT+/HER2+ tumors in the 2009 study were all Elston grade 3; two of three had a bleak prognosis [73]. These preliminary findings suggest that a certain subtype of HER2+ breast cancers may be more prone to activate the ALT pathway. Large-scale studies to confirm this relationship would have important prognostic and therapeutic value.

Breast carcinomas with metastatic potential tend to have a poor prognosis [75]; however, the mechanisms that confer this potential are poorly understood. An interesting report by Robinson and colleagues found that poor prognosis of HER2+ breast cancers may be associated with increased expression of the SLX4 interacting protein, or SLX4IP [76]. The SLX4-containing structure-specific endonuclease (SSE) complex plays an important role in the ALT pathway, working to resolve recombination intermediates and serving as a competitor to the BLM-dependent dissolution pathway [2]; SLX4IP is also a member of the SLX4 SSE complex [76]. In the D2.0R cells, an ALT+ murine mammary cancer cell line, knockout of SLX4IP resulted in upregulated hTERT, decreased APBs and C-circles, and re-expressed ATRX and DAXX [76]. Ectopic re-expression of SLX4IP reversed the induced TEL+ phenotype back to ALT+. Thus, SLX4IP appears to be a positive regulator of ALT in the D2.0R cells. These findings appear to contrast results observed in OS; wherein SLX4IP loss has been associated with an increase in ALT-related phenotypes [77].

Based on these in vitro results, the investigators then examined the expression level of SLX4IP and hTERT in a large cohort of human breast cancer samples. They found correlations between SLX4IP/hTERT expression and prognosis [76]. For patients with TNBC, those with SLX4IP^high^/hTERT^low^ had a better prognosis than patients with SLX4IP^low^/hTERT^high^. However, for patients with HER2+ breast cancer, those with SLX4IP^high^/ hTERT^low^ had a worse prognosis than patients with SLX4IP^low^/hTERT^high^, which is consistent with the findings from the 2009 and 2013 studies on HER2 and ALT positivity discussed above [73,74]. Again, an ALT molecular phenotype confers a worse prognosis in HER2+ tumors. Additionally, it appears that there may not only be a subset of HER2+ breast cancers that are ALT+ but there may also be some TNBC that exhibit ALT activity.

Two ATR inhibitors were found to induce cytotoxicity and reduce the growth of murine and human breast cancer cell lines that were SLX4IP^high^/hTERT^low^ (ALT+) [76]. This effect was ALT specific as ATR inhibition was less effective against the SLX4IP^low^ (ALT−) cells, regardless of hTERT mRNA status. Reciprocally, the telomerase inhibitor, 5-fluorouridine (5-FU), was more effective against the SLX4IP^low^/hTERT^high^ (TEL+) cells. Interestingly, however, human breast cancer cells seem to be capable of modulating the expression of SLX4IP to switch TMMs to become chemoresistant [76]. For example, prolonged treatment of the TEL+ human TNBC cell line, HCC1806 (SLX4IP^low^/hTERT^high^), with 5-FU, though initially effective, resulted in the upregulation of SLX4IP, increased APBs, resistance to 5-FU, and more sensitivity to ATR inhibitor [76]. An alternative explanation is the co-existence of both TEL+ and ALT+ within the HCC1806 cells, to begin with. These findings suggest the importance of adopting dual treatment strategies that target both ALT and TEL activity to maximize therapeutic efficacy.

ALT activity was also observed in mouse tumor-derived mammary stem cells (MSCs) recently [78]. Using TRF2-deficient mouse models, Wu and colleagues observed significantly greater C-circles, APBs, and telomeric recombination events in MSC-originated tumors than in luminal progenitor (LP) originated tumors. MSC tumors also manifest greater tumorigenicity than LP tumors [78]. Although the hierarchy of mammary stem cell lineage has not yet been clearly elucidated, these findings postulate that ALT activity may have unexplored embryologic connections and ALT inhibitors may be more effective in the treatment of MSC-originated cancers.

In summary, a small percentage of HER2+ breast cancers may rely on the ALT pathway to maintain their telomeres. The addition of an ALT-targeted therapy to the existing treatment regimen will likely improve the chance of survival for these breast cancer patients.

## 4. ALT Positive Central Nervous System (CNS) Tumors

### 4.1. Gliomas

Gliomas are the most common primary brain tumors in adults [79,80]. The prevalence of ALT in gliomas (studies with mixed oligodendroglial and astrocytic tumors or unspecified gliomas) ranges from 13 to 69% (Table 2) [40,81,82,83,84,85,86,87,88,89,90,91]. Further subdivision of gliomas based on their cell origin, histology, and genetic alterations have revealed that the ALT phenotype varies among tumor types. TMM is commonly assessed in astrocytomas, with ALT measuring between 10% and 78% overall (Table 2). Astrocytomas are further broken down by the World Health Organization (WHO) into grade I–IV, with grade IV astrocytoma being synonymous with glioblastoma multiforme (GBM) [92]. Subgroup analysis reveals that the ALT phenotype is most prevalent among grade II and III astrocytomas, at about 55% and 65%, respectively (Table 2). Henson and colleagues first observed the ALT phenotype in a significantly higher number of grade II/III astrocytomas compared to their grade IV counterparts, present in 88% and 24% of tumors, respectively [38]. Similarly, a later study showed that total telomere length and the expression of a telomeric long noncoding RNA, TERRA, were inversely related to astrocytoma grade [93]. Furthermore, ALT+ tumors had 12-fold and 2-fold higher TERRA expression than TEL+ tumors and tumors lacking known TMM, suggesting that ALT may indeed play a more pronounced role in lower grade astrocytomas [93]. Consistent with this finding, TERRA was shown later to enhance the recombinogenic nature of ALT telomeres [94].

ALT is relatively rare in grade I astrocytomas with the highest reported rate of 3% [40,95]; however, ALT was present in 69% of tumors in an anaplastic subtype found in patients with neurofibromatosis type I (NF1) [96]. This suggests that genetics may strongly influence the choice of TMM. Interestingly, grade I astrocytomas were found to be positive with APBs but lacked long and heterogeneous telomeres as measured by TRF, which prompted Slatter and colleagues to classify grade I astrocytomas in their study as “telomere-associated promyelocytic leukemia (PML) bodies” positive and yet ALT negative [95]. Whether this unique phenotype potentiates ALT induction or tumor stage progression remains unclear.

In GBM, the most aggressive type of glioma, the presence of ALT varies greatly between 7% and 47% [81,85,95,97,98,99,100,101,102,103,104,105,106]. In large cohorts of adult glioblastoma, the prevalence of ALT is approximately 16%, with a range of 11 to 25% (Table 2). Pediatric glioblastoma measures a much higher prevalence at about 39% (Table 2).

Heaphy and colleagues first reported ALT in oligodendrogliomas at a rate of 20% (8/40) [40]. Others have reported the prevalence of ALT in oligodendrogliomas as completely absent to 25% in smaller cohorts, all using the telo-FISH assay [81,83,84,85]. Combined analysis indicates the prevalence of ALT in grade II/III oligodendrogliomas to be around 17% (13/78) [40,81,83,84,85]. On the other hand, ALT is present in about 60% (29/48) of grade II/III oligoastrocytomas, which, together with genetic data, suggests a clear predominance of astrocytic cells in comparison to oligodendroglial cells in mixed-lineage tumors [83,84,85]. Finally, in ependymoma, another subtype of glioma, ALT is completely lacking (Table 2).

Taken together, it is quite clear that the prevalence of ALT in gliomas is highly dependent on their cellular origins. ALT is particularly prevalent in grade II and grade III astrocytomas but is less prevalent in glioblastoma. Though less comprehensively studied, ALT appears less prevalent in other glioma subtypes, such as oligodendrogliomas and ependymomas.

#### 4.1.1. Genetic Alterations, Cellular Lineage, and Survival in ALT Positive Gliomas

Loss of ATRX expression and the presence of ALT are significantly correlated in gliomas (Table 3) [81,82,84,91,104,105]. In an N-Ras mutant and p53-null pediatric glioblastoma mouse model, knockdown of ATRX induced the ALT phenotype, as measured by telo-FISH and CCA, in a subset of tumors [107]. Staining of ATRX-null tumors revealed a loss of DNA-PK at the DNA damage sites and impaired non-homologous end joining (NHEJ). Thus, Koschmann and colleagues proposed that the inactivation of ATRX shifts the repair of DSBs from NHEJ to HR, which potentiates the induction of ALT [107]. ATRX loss alone, however, is insufficient to induce ALT since knockdown of ATRX in two TEL+ human glioma cell lines, 8-MG-BA and Hs-683, did not increase the formation of C-circles or APBs [108]. Intriguingly, knockout of ATRX in four TEL+ high-grade gliomas (HGG) cell lines led to an increase in APBs and C-circles in half of the cell lines while they still retained telomerase activity [109]. These in vitro findings suggest that high-grade gliomas may be capable of maintaining telomeres by both TMM simultaneously.

Other frequent genetic alterations that occur in ALT+ gliomas involve *TP53* [104,110] and *IDH* [82,98,104,105]. These mutations correlate with alterations in ATRX [81,82,83,84,85,91,111]. Mukherjee and colleagues showed that the combination of ATRX loss and IDH1 R132 mutation might be sufficient for ALT activation in gliomas [112]. TP53-null, Rb-null, IDH1 R132H mutant human astrocytes expressing WT ATRX reactivated the hTERT promoter after prolonged proliferation while their ATRX knockout counterparts activated the ALT pathway. Although both telomerase and ALT counterparts had reduced expression of RAP1 and XRCC1, knockdown of RAP1 and XRCC1 together with ATRX loss increased C-circles, APBs, and tSCE similar to the ATRX-null and IDH1-mutant astrocytes. They thus concluded that, in the context of ATRX loss, IDH1 R132H inhibits RAP1 and XRCC1, which leads to telomeric dysfunction and impaired alternative NHEJ, respectively, and thus promotes HR and ALT. The direct role of p53 in ALT induction in gliomas is unknown. Kannan and colleagues suggest that the loss of p53 may be to avoid apoptosis in the presence of genomic instability due to ATRX loss and enhanced stem cell proliferation due to IDH mutations [84].

Many genetic alterations are diagnostic biomarkers in gliomas. ATRX loss does not co-exist with 1p/19q co-deletions [84,91,113], while 1p/19q co-deletions are correlated with *IDH* mutations [82]. In addition, *CIC* and *FUBP1* mutations are associated with 1p/19q co-deletion, *IDH* mutations, and oligodendroglial cells and were mutually exclusive with *TP53* and *ATRX* mutations [83]. In fact, the co-occurrence of ATRX loss, *TP53* mutations, and IDH1 R132H are restricted to the astrocytic lineage [81,83,84,85,91,114]. ATRX retention, 1p/19q co-deletion, and *IDH1* mutations are limited to the oligodendrocytic lineage [81,84,85,91,113,114]. These studies, amongst others, likely led to the reclassification of gliomas in the WHO 2016 classification of CNS tumors [92]. Lastly, ATRX loss and *hTERT* mutations rarely occur together in gliomas [105,114], and hTERT alterations are much more prevalent in the oligodendrocytic lineage [114]. Thus, astrocytomas primarily utilize ALT while oligodendrogliomas favor telomerase for telomere maintenance.

In infantile and pediatric gliomas, a similar association between the ALT phenotype and ATRX loss has been observed [66,88,89,107,115]. ALT is also associated with *TP53* mutations in diffuse intrinsic pontine glioma (DIPG) and HGG [66,88,116]. Unique to pediatric gliomas are mutations in histone H3.3. Two studies have reported that 100% of tumors with H3.3 G34R/V mutations were ALT+ [89,117]. Another study reported that all 13 tumors with H3.3 G34R/V mutations had a concomitant loss of ATRX, which was highly correlated with ALT [66]. On the other hand, 40–50% of H3.3 K27M mutant tumors exhibited ALT, and only 38.5% (5/13) H3.3 K27M-mutant glioblastomas were also ATRX-null [66,89,117]. In addition to *H3.3* and *IDH*, mutations in the *SETD2* gene, which encodes the only H3K36 trimethyltransferase in humans, have also been identified in 15% of pediatric HGGs and 8% of adult HGGs [118]. Most intriguingly, mutations in the *SETD2* gene are mutually exclusive with the H3.3 mutation, suggesting that SEDT2 and H3.3 function in the same biological process. Though the TMM status was not examined in the *SETD2* mutant HGGs, based on the intimate connection of H3.3 mutated HGGs with ALT+, one would predict that the SETD2 mutant HGGs mostly rely on the ALT pathway as well. Collectively, these data suggest that histone mutations may be independent or synergistic inducers of ALT in pediatric DIPG and HGG. ALT, in the presence of *TP53* mutation, was found to significantly improve overall survival in pediatric DIPG and HGG [88], while two other studies did not find a relationship between ALT and survival [115,117].

Patients with *NF1* mutations are susceptible to CNS tumors. Gliomas in patients with NF1 loss have a prevalence of ALT ranging between 29% and 69% (Table 2). The ALT phenotype co-exists with loss of ATRX expression, especially in HGGs [86,87,90]. This suggests that ATRX loss in the presence of *NF1* mutation may be sufficient to induce ALT in gliomas. Interestingly, IDH, histone H3, and hTERT alterations were rare in ALT+, ATRX-null, and *NF1*-mutant gliomas [86,90,96,119], while the co-occurrence of *TP53* mutations remains unclear [86,90,119]. Copy number loss and deletions in *CDKN2A* and *CDKN2B* have been observed in ALT+, ATRX-null HGG, suggesting the existence of other genetic drivers for ALT induction in NF1-deficient tumors [87,90]. Furthermore, sequencing of three subependymal giant cell astrocytoma (SEGA)-like astrocytomas in *NF1* mutant patients with ALT and intact ATRX have revealed genetic alterations in RECQL4, Fanconi anemia complementation group genes (FANCD2, FANCF, and FANCG), and SMARCAL1 [119]. Lastly, *NF1* mutant patients with ALT+ gliomas had significantly worse survival than ALT− gliomas with elongated or normal telomeres [87].

Recent studies have identified novel genetic alterations in ALT+ gliomas. Foremost, in pediatric and adult HGG, the presence of ALT correlated with *PDGFRA* amplification [115], a gene with copy number variations in 8% of WT *hTERT* and *IDH* glioblastoma [100]. The mechanism by which PGFRA may contribute to ALT activation requires further investigation. Secondly, in WT hTERT and IDH glioblastoma, all ALT+ tumors have either ATRX loss or SMARCAL1 mutation [100]. SMARCAL1 is thought to alleviate replication stress by reversing stalled replication forks for repair, thereby opposing the ALT mechanism [58]. D06MG, a glioblastoma cell line with *SMARCAL1* nonsense mutation and intact *ATRX*, was ALT+ as measured by APBs and C-circles. Overexpression of SMARCAL1 in D06MG reduced ALT activity as indicated by fewer ultra-bright telomeric foci. SMARCAL1 knockout in ALT− glioblastoma cells, U87MG and U251MG, significantly increased C-circles and APBs. Thus, the inactivation of SMARCAL1 facilitates the induction of ALT. Finally, sequencing of paired patient-derived normal and glioblastoma tissue revealed elevated expression of Mucin 1 (MUC1) in tumors [120]. Knockdown of MUC1 in glioblastoma cell lines, U373 and T98G, decreased proliferation and induced cell cycle arrest. In addition, hTERT expression and telomerase activity were significantly decreased, and C-circle formation significantly increased along with a slight increase in telomere length. Silencing of MUC1 is sufficient to switch glioma cells from TEL+ to ALT+.

ALT+ is often associated with younger age at diagnosis [38,82,83,97,98,121] and better survival [82,97,101,104,105,110] compared to ALT− gliomas (Figure 1). Furthermore, ATRX loss [105,111], *IDH* mutations [82,105,111], and *TP53* mutations [110] enhanced survival compared to their WT counterparts. Patients with *NF1* mutations appear susceptible to ALT+ CNS tumors with poor prognosis [87,96]. The genetics of ALT positivity and gliomas is rapidly developing and promising for the development of better management strategies.

#### 4.1.2. Treatment for ALT Positive Gliomas

The current standard of care for HGG includes surgical resection, radiotherapy, and chemotherapy. Despite improved survival of patients, disease outcome is still poor and relapse is very common [122]. Effective therapeutics are urgently needed. A limited number of studies evaluated the treatment of ALT+ versus TEL+ gliomas. In the *N-Ras* mutant and *TP53*-null pediatric glioblastoma mouse model, compared to ATRX-WT tumors, inactivation of ATRX made tumors more susceptible to DSBs generated from radiation and chemotherapies, but not SSBs [107]. This suggests that ATRX-null tumors may be more susceptible to DSB-inducing agents due to impaired NHEJ. In addition, ATRX-null glioma cells are more susceptible to temozolomide than ATRX-WT [118]. Thus, DSB-inducing agents may have promising effects in treating ALT+ gliomas.

### 4.2. Other CNS Tumors

In pediatric patients, ALT was present in 7 out of 31 choroid plexus carcinomas (CPCs) at a prevalence of 23% (Table 2) but was absent in the benign choroid plexus papillomas [88]. All ALT+ CPCs were also deficient in p53. In the *TP53*-null CPCs, ALT positivity showed a trend for improved survival compared to those that are ALT−.

ALT+ medulloblastoma is estimated to be about 7% prevalent (Table 2) [40,88]. Intriguingly, when surveying 43 pediatric metastatic medulloblastoma, Minasi and colleagues recently found that 30% of tumors showed negative ATRX nuclear staining, suggesting that ALT positivity in medulloblastoma may be higher than previously reported [123].

## 5. ALT Positive Neuroendocrine Tumors

### 5.1. Neuroblastoma

Neuroblastoma (NB) is a malignant tumor of the sympathetic nervous system that most commonly occurs in pediatric patients. It is of neuroectodermal origin, arising from the embryonic neural crest, and is the most common extracranial solid tumor of childhood and the most frequently diagnosed cancer of infancy [124,125]. The pathophysiology of NB is variable and can present as an asymptomatic mass, locally invasive, or even disseminated disease [125]. The risk of NB can be assessed through tumor stage, patient age, histology, as well as MYCN amplification [126,127]. The survival rate of high-risk NB patients is below 50% [128].

ALT prevalence in NB has been estimated to fall between 9% and 59% (Appendix A). This is a very broad and eclectic range, attributable to a predominance of small sample size in many studies. Surveying all available data narrows this range to 18–47%, with an average of 24% manifesting the ALT phenotype (Table 2).

Early TMM studies in NB suggested that ALT+ and TEL+ NB are both correlated with high-risk tumor status, later age of onset (>18 months), poor histology, and poor clinical outcomes [129,130]. However, three recent systematic NB studies compared ALT-related and TEL-related outcomes and found the presence of TEL to be less favorable than ALT (Figure 1) [126,127,128]. The 5-year overall survival rate of TEL+/ALT− high-risk NB and TEL−/ALT+ high-risk NB are 28% and 46%, respectively [126]. In contrast, the 5-year overall survival rate of TEL−/ALT− is 75%. Intriguingly, one study found that 33% (20/60) of ALT+ NB appeared to concomitantly exhibit a high level of hTERT (ALT+/TEL+) and had the worst prognosis [127].

#### 5.1.1. ALT Positive Neuroblastoma and the Mutation Status of ATRX/DAXX

Loss of function ATRX mutations have been observed in 8% (72/898) of NB tumors overall (Table 3). ALT+ NB experiences loss of ATRX at a frequency of 78%, but if ATRX is lost, NB is ALT+ 92% of the time. ATRX aberrations are also associated with poor prognosis, high-risk NB, and later age of diagnosis [131,132,133]. Though it has been frequently screened for, *DAXX* mutations are rare in NB (Table 3). To our knowledge, only one such case has been observed so far [132].

#### 5.1.2. MYCN Amplification and ALT Positivity Are Mutually Exclusive and Synthetically Lethal

MYCN normally functions as a transcription factor during embryonic development and has been implicated in NB pathophysiology as early as 1997 [134]. Overexpression of MYCN promotes the generation of reactive oxygen species (ROS), metabolic reprogramming, mitochondrial dysfunction, and DNA replication stress [135]. MYCN amplification is present in 20–30% of NB and portends a poor prognosis. It is also the most important genetic abnormality used in the risk stratification of NB [136].

So far, at least nine independent studies have reported that MYCN amplification in human NB rarely overlaps with ALT positivity or ATRX-mutation [126,128,130,131,132,133,135,137,138]. For example, in the first characterization of four ALT+ NB cell lines, all lacked MYCN amplification [138]. Early clinical analysis of this phenomenon by Lundberg and colleagues found that poor patient survival was observed in TEL+ NB with MYCN amplification and in ALT+ NB without MYCN amplification [137]. In contrast, the TEL+ NB without MYCN amplification had better cumulative survival. MYCN amplification results in extensive methylation of the *hTERT* gene and its promoter, resulting in elevated telomerase activity [126,133,139]. Additionally, genomic rearrangements near *hTERT*, prevalent in 24% of high-risk NB, also elevate telomerase activity and independently confer poor prognosis [133]. Interestingly, inhibition of telomerase activity in NB cell lines by 6-thio-dG demonstrated that *hTERT*-rearranged NB cells are more susceptible to cytotoxicity than MYCN amplified cells [128].

Notably, *hTERT* rearrangements occur mutually exclusive with MYCN amplification and ATRX loss [133]. The overall survival rate of MYCN amplified, *hTERT*-rearranged, and ALT+ NB is about 52%, 42%, and 78%, respectively [128]. If none of these three aberrations are present, the overall survival rate increases to 92% [128]. Considering MYCN-amplification and *hTERT*-rearrangements both result in elevated telomerase activity and worse prognosis than ALT+, TEL and ALT activity may be more useful as prognostic indicators than MYCN amplification alone.

MYCN overexpression and ATRX loss appear to be synthetically lethal, which helps to explain the observation mentioned above that MYCN amplification and ATRX mutation are mutually exclusive [135]. In a recent study, Zeineldin and colleagues demonstrated that ATRX knockout led to decreased colony formation in MYCN amplified NB cell lines, while no change was observed in MYCN WT NB cell lines [135]. Correspondingly, inducing MYCN expression in ATRX-mutant NB cells and in U2OS cells (ATRX-mutant) also resulted in significant loss of viability. In mouse in vivo growth competition assays, MYCN-induced SKNMM NB cells (ATRX-mutant) were outcompeted by the non-induced SKNMM cells [135]. Electron microscopy studies showed that concomitant ATRX mutant and MYCN amplification resulted in mitochondrial disruption [135].

Though the molecular mechanism remains unclear, the synthetic lethality of MYCN amplification and ATRX mutation might be explored as a novel therapeutic strategy for treating NB patients.

### 5.2. Pancreatic Neuroendocrine Tumors (PanNETs)

Pancreatic Neuroendocrine Tumors (PanNETs), also known as islet cell tumors, comprise 1–2% of all pancreatic tumors and are thought to derive from cells of pluripotent pancreatic cells of the ductal/acinar system [140]. PanNETs can be divided into two groups according to secretory status: functional and non-functional. Most PanNETs have slow, indolent growth and are asymptomatic. Therefore, a majority of patients with PanNET are at an advanced stage at diagnosis [140]. PanNETs are rare tumors with multifaceted presentations that continue to confound the medical community; thus, a consensus on the standard of care has yet to be met.

Relative to ALT− cases, ALT+ PanNETs are associated with larger tumor size [141,142,143,144,145], higher grade [142,144,145,146], greater lymphovascular invasion [143,144,145], more chromosomal gains [146], and greater risk of metastasis [143,144,145,147]. Composite APBs and telo-FISH biomarker data from our large cohort of surveyed studies estimate PanNET ALT positivity at about 32% (*n* = 1152) (Table 2).

#### 5.2.1. PanNET and the Mutation Status of MEN1 and ATRX/ DAXX

Loss of function of ATRX/DAXX has been observed in 32% of PanNETs (394/1223). ALT+ PanNETs have ATRX/DAXX aberrations 86% of the time (Table 3). However, if ATRX/DAXX is lost, PanNETs are ALT+ in 96% of cases. The ALT phenotype, as well as loss of ATRX/DAXX expression, are associated with poor prognosis in patients with PanNETs (Figure 1) [141,144,145,148,149,150], while only one report found the opposite [151]. Intriguingly, subgroup analysis has repeatedly demonstrated that ALT+ metastatic PanNETs have better overall survival than those that are ALT− or ATRX WT [65,145,151,152].

In other ALT+ cancers, ATRX mutations significantly predominate DAXX mutations (Table 3). However, with prevalence at about 11%, PanNET patients have an unusually higher percentage of tumors that bear DAXX aberrations. The reason for this remains unclear. Since ATRX and DAXX function together to remodel chromatin, it was thought that a mutation in either of these genes would be sufficient to enhance ALT activity. Somewhat unexpectedly, at least 19 PanNET cases have been observed to lose the function of both genes (Table 3), suggesting that ATRX and DAXX may have a non-overlapping function during the development of PanNETs.

*MEN1*, the gene on chromosome 11q whose germline mutation results in MEN1 syndrome, is also the most frequently mutated gene in sporadic PanNETs [153]. In both sporadic and syndromic PanNETs, *MEN1* functions as a tumor suppressor gene with loss of function of both alleles (either by germline mutations, somatic intragenic mutation, or loss of heterozygosity). About 44% of PanNETs have MEN1 aberrations, while about a quarter of PanNETs have both ATRX/DAXX and MEN1 loss [65]. Recent evidence suggests that PanNETs with *ATRX*, *DAXX*, and/or *MEN1* mutations are either alpha-islet-cell in origin or differentiate to become genetically similar to alpha-islet-cells [150,154,155]. Moreover, those that exhibit this phenotype experience more tumor recurrence [150,154,155]. DNA and RNA sequencing, as well as methylation analysis, indicates that PanNETs can also be separated into two cohorts: ARX+/PDX1- and ARX-/PDX1+ [150,154,155]. ARX (aristaless related homeobox) and PDX1 (pancreatic and duodenal homeobox 1) are pancreatic alpha-cell and beta-cell biomarkers, respectively. As with ALT+ status, ARX+ status independently predicts worse recurrence-free survival [150,154,155]. When ALT and ARX were evaluated together, remarkably, relapse was found in every ARX+/ALT+ tumor, while it occurred less frequently in all other subgroups [154]. Initial investigations primarily focused on the non-functional PanNETs, as they predominate and are more likely to metastasize [156]. However, a 2020 report confirmed that when metastatic, functional PanNETs, particularly insulinomas, also resemble alpha-cells and are ARX+/ALT+ [155]. Therefore, ARX positivity, mutations in the *ATRX*, *DAXX*, and *MEN1* genes, and ALT positivity seem to be all interconnected to the pathophysiology of metastatic PanNETs.

Recent research suggests that *MEN1* mutations occur early in the development of PanNET, while ATRX/DAXX loss and the ALT phenotype may be late changes [157,158]. Pancreatic microadenomas are thought to be precursor lesions to PanNETs [157]. In two independent studies, no ATRX/DAXX loss nor ALT activity was observed in both MEN-1 syndrome microadenomas and sporadic microadenomas (0/47 and 0/19 samples, respectively) [157,158]. Meanwhile, all MEN1-syndrome cases and 14 out of the 19 sporadic microadenomas exhibited aberrant MEN1 activity. The absence of ALT activity in microadenomas suggests that the ALT phenotype may be a later development in PanNET tumorigenesis.

#### 5.2.2. PanNET Diagnosis

The associations of all these factors with prognosis and metastasis atop the technical ease of using any of these genes as biomarkers strongly support a change in testing guidelines of PanNETs. MEN1, ARX, ATRX, DAXX, and ALT should all be considered in future PanNET genetic and histopathological assessments.

In light of ALT positivity conferring worse prognosis in PanNETs, recent research has validated means of identifying these tumors earlier and easier. Two recent studies have affirmed that fine-needle aspiration (FNA) biopsies, generally short outpatient procedures, are nearly as effective as excisional biopsies at determining the ALT and ARX status in PanNETs [142,147]. From abdominal CT scans of patients later confirmed to have PanNETs, another group demonstrated that the presence of pancreatic duct dilation, hepatic metastasis, and tumor size larger than 3 cm, together, were highly predictive of ALT+ tumor status [143]. The absence thereof was predictive of ALT− status. Reliable and specific, these studies are the first to highlight plausible methods of acquiring the important prognostic information earlier. The use of these non-invasive methods could provide evidence to support or expedite definitive excision and/or pathological characterization.

In about 50% of cases, at first presentation, PanNETs have already metastasized to the liver; and unfortunately, liver metastases are the leading cause of death in PanNET patients [147]. Furthermore, the primary sites of non-functional PanNET metastases are occult to routine radiological screening 46% of the time [159]. As neuroendocrine tumors can arise in many tissues, including the lung, thyroid, pancreas, and tubular gastrointestinal tract, this is not surprising. In addition to the pathological assessment of the excised metastasis, exploratory laparotomy may still be required to locate the primary tumor for excision [159]. ALT positivity in neuroendocrine liver metastases is 96% specific at indicating that the primary tumors are of PanNET origin [152]. Therefore, assaying liver metastases for ARX or ALT biomarkers via previously discussed FNA methods would facilitate locating and diagnosing pancreatic origin lesions. If an ALT+/ARX+ tumor is identified in the liver, this would not only indicate early that the unknown primary lesion is a PanNET but also that it is likely to be amenable to surgery.

## 6. ALT Positive Soft Tissue Tumors

### 6.1. Angiosarcoma

Angiosarcomas are rare, aggressive, vascular tumors of mesenchymal cell origin that can arise from many sites within the body but commonly present as a cutaneous disease of the head and neck in elderly white males [160]. Patients often have histories of radiation therapy, chronic lymphedema, or exposure to certain chemicals, such as vinyl chloride [160]. Angiosarcomas characteristically have a poor prognosis [161]. Pathogenetically, they are heterogeneous tumors with a range of varying genetic events, though many genetic changes appear linked to the MAPK signaling pathway [162]. Recent studies showed that angiosarcomas are highly correlated with the ALT phenotype and ATRX loss, with ALT being prevalent in 11–24% of angiosarcomas [40,163] and ATRX loss in 6–18% of angiosarcomas [161,163].

In a 2015 study conducted by Liau and colleagues, 24% (17/70) of angiosarcoma tumors were shown to be ALT+ [163]. Interestingly, all seventeen samples were primary angiosarcomas [163]. Furthermore, Liau and colleagues also observed ALT significantly more often in hepatic angiosarcomas than in those that were non-hepatic primary angiosarcomas [163]. ATRX loss was observed in 18% (16/88) of the angiosarcomas. Similarly, all ATRX-deficient angiosarcomas were primary angiosarcomas, with hepatic angiosarcomas having the highest rate of ATRX loss relative to non-hepatic tumors [163].

The findings by Liau and colleagues suggest that the ALT phenotype and ATRX loss are highly correlated with angiosarcomas, especially the primary hepatic angiosarcomas. These findings may be useful in the understanding of angiosarcoma prognosis. In a cohort of 118 angiosarcoma cases, Panse and colleagues found 6% (7/118) to have ATRX loss, and like the findings of Liau and colleagues, they were all primary angiosarcoma [161]. Intriguingly, all secondary angiosarcomas retained expression of ATRX, suggesting that ATRX loss is a characteristic of primary tumors only [161]. In addition, they found that angiosarcoma patients with ATRX-mutant had decreased event-free survival compared to patients that retained ATRX expression (Figure 1).

### 6.2. Leiomyosarcoma

Leiomyosarcoma (LMS) is a malignant tumor of mesenchymal origin and is considered one of the most frequent soft tissue sarcomas, having an incidence rate of 10–20% of all newly diagnosed soft tissue sarcomas [164,165]. LMS can arise in many soft tissue sites, such as the retroperitoneum and large blood vessels, but more commonly, it arises from mesodermal tissue (smooth muscle) of the uterine wall [164]. Diagnosing uterine leiomyosarcoma (ULMS) can be difficult; they may sometimes be mistakenly diagnosed as uterine leiomyoma (benign uterine fibroids) [166]. Finding genetic contributions that are characteristic of LMS will aid the earlier and proper diagnosis and treatment of LMS.

Previous studies have shown that the prevalence of ALT+ LMS ranges from 42% to 78%. Surveying of all the currently available data narrows the range between 52% and 78%, depending on the ALT biomarker used (Table 2 and Appendix A). On average, ALT is observed in approximately 60% of LMS cases (*n* = 331), implicating LMS high on the list of ALT-associated malignancies.

ALT positivity in ULMS is correlated with increased tumor size and high mitotic index [167], suggesting that ALT+ ULMS may be more aggressive and lead to a poorer prognosis. Correspondingly, ALT+ ULMS patients were found to have shorter disease-free survival and overall survival rates than the ALT− ULMS patients (Figure 1) [167]. These findings support similar poor ALT+ prognosis data observed in uterine sarcomas broadly [168].

#### ALT+ Leiomyosarcoma and the Mutation Status of ATRX/DAXX

Studies have found that ATRX loss is also associated with poor LMS patient prognosis. ATRX was first implicated in LMS in three 2015 publications [167,169,170]. *ATRX/DAXX* mutations have been observed in 34% (103/311) of all tested LMS tumors (Table 3). ALT+ LMS experienced ATRX loss 56% of the time; however, if ATRX is lost, LMS was ALT+ 83% of the time. After assessing tumor pathology, *ATRX* mutations correlated with poor differentiation status, tumor necrosis, and the ALT mechanism [167,169]. Slatter and colleagues presented similar findings: 23/26 (88.4%) ULMS patients who died or had recurrent disease were found to have ATRX or DAXX loss in their tumors [170]. In contrast, the three patients who did not harbor ATRX or DAXX aberrations showed no disease in periodic follow-up between 59 and 156 months [170]. There is likely a strong correlation between ATRX/DAXX aberration and poor LMS prognosis; however, the small sample size has limited the use or interpretation of Kaplan–Meier survival comparisons [170,171]. Interestingly, ULMS showed a trend of more frequent ATRX loss compared to non-uterine LMS sites (18/46 vs. 2/16), indicating a strong and specific correlation between ATRX loss and ULMS [167].

Uterine smooth muscle tumors of uncertain malignant potential (STUMP) are rare malignancies defined by histological features that make them undefinable as benign or malignant, which often leads to misdiagnosis [172,173]. Since there is a correlation between ATRX/DAXX loss and an ALT+ phenotype in ULMS [167,170], these characteristics could potentially be used as markers to better discern STUMP entities as either uterine leiomyomas or leiomyosarcomas. Slatter and colleagues found that 100% of the STUMP patients examined (*n* = 6), who had died or had recurrent disease, harbored ATRX or DAXX loss in their tumors [170]. These paralleled the findings observed in ULMS [167,170]. Interestingly, the ALT phenotype is rarely found in leiomyomas (benign uterine tumors) (Table 2), suggesting that ALT may be specific to ULMS and STUMP, and therefore could be used as a diagnostic biomarker for the more aggressive uterine tumors. These findings are important and merit follow-up as ULMS and STUMP are often misdiagnosed for uterine leiomyoma due to overlapping symptoms and/or histology [166,172].

Though ALT is absent in conventional uterine leiomyoma (UL), recent evidence suggests that certain UL subtypes, particularly leiomyoma with bizarre nuclei, have greater potential to exhibit ALT activity and evolve into more aggressive lesions [166]. In a cohort of variant uterine leiomyoma patient samples, 6% (9/142) showed ALT-relevant aberrations, such as ATRX/DAXX loss and/or ultra-bright telomeric foci [166]. No conventional ULs (0/64) displayed such aberrations. Two of the nine patients with ALT-related tumors presented later with benign metastasizing leiomyoma to the lungs. Further, though infrequently arising in benign tumors, aberrations appear to be unique to variant UL [166]. Interestingly, the variant UL, with bizarre nuclei, showed the most frequent ALT-relevant aberrations (5/31, 16%). Unlike the other variant UL histological subtypes, this subtype is thought to have a similar mutation profile to ULMS, which includes *FH* inactivation, infrequent *MED12* mutation, and *TP53* mutation [166]. This subtype also recurs more frequently than other UL subtypes [166]. Variant ULs with bizarre nuclei have characteristics that differentiate them from other variant ULs. Furthermore, those that harbor ALT aberrations share both genetic and histological traits similar to ULMS. Therefore, ALT biomarkers will likely be beneficial diagnostic tools for ULMS and the variant UL with bizarre nuclei, leading to better diagnosis and management.

### 6.3. Liposarcoma

Liposarcoma (LPS) is a malignant group of soft tissue sarcomas derived from adipose tissue, which accounts for about 20% of all mesenchymal cancers and most commonly develops in the retroperitoneum and extremities [174,175]. LPS can be divided into four histologic subtypes: well-differentiated, de-differentiated, myxoid, and pleomorphic [174,175]. Well-differentiated and de-differentiated LPS account for 40–50% and 15–20% of all LPS, respectively [176]. Myxoid LPS accounts for 20–30% of cases, while pleomorphic LPS accounts for 5–10% [176]. Well-differentiated LPS generally have no metastatic potential; however, they can de-differentiate if left untreated [176]. Pleomorphic LPS is the most aggressive LPS subtype [176].

Studies have shown that in LPS, ALT positivity ranges between 25% and 29% (Table 2). In LPS, ALT activity is correlated with increased patient age, high-grade tumors, high mitotic count, and the presence of necrosis [1,2,3,4]. It is estimated that 44% of grade III LPS are ALT+ while 16% are TEL+ and 36% are ALT−/TEL− [174]. Clinically, ALT+ LPS is associated with worse prognosis (Figure 1) [174,177,178,179]. Costa and colleagues were the first to systematically investigate the LPS prognosis in a large cohort (*n* = 139) [174]. They found that ALT−/TEL+ LPS had a hazard ratio of 2.58 while ALT+/TEL− LPS had a hazard ratio of 6.39 at 10-years [174]. Another study by Venturini and colleagues demonstrated similar results: after 10-years, only 45.5% of ALT+ LPS subjects survived, relative to 71.1% of ALT− subjects [178]. The trend for the survival rate remained true at 15 years, which was 25.3% and 71.1% for ALT+ and ALT− LPS patients, respectively [178]. Interestingly, Cairney and colleagues determined that the presence of ALT and increased hTR expression were associated with poor prognosis [179]. Taken together, these results suggest that the increased mortality in LPS is highly associated with ALT positivity.

Aggressive LPS subtypes appear to activate the ALT pathway more frequently than telomerase. For example, between 30% and 78% of de-differentiated LPS are ALT+ [174,179,180], while only 18% are TEL+ [174]. Conversely, ALT positivity in well-differentiated LPS is much less frequent, ranging from 0% to 14% [174,179,180]. Usual myxoid LPS, a less aggressive LPS, is ALT+ about 13% of the time and TEL+ 39% of the time [174]. Recent preliminary data from the only published assessment of ALT prevalence in pleomorphic LPS, the most aggressive and metastatic subtype, found evidence of ALT positivity in 80% of cases (*n* = 10) [180]. This same study found that none of the well-differentiated LPS samples (*n* = 16) were ALT+ [180]. Taken together, evidence strongly suggests that ALT activity is more likely implicated in aggressive LPS subtypes than telomerase. Round-cell LPS, a notably aggressive myxoid LPS subtype, appears to be the exception, among which 69% are TEL+, and only 15% are ALT+ (*n* = 26) [174].

Interestingly, while ALT+ LPS tend to be more aggressive, TEL+ LPS are more prone to metastasize [174]. In both primary and recurrent lesions, the frequency of ALT positivity and TEL positivity ranges from 19–25%. However, in metastatic disease, while ALT is only 18% prevalent, TEL is 59% prevalent [174].

#### 6.3.1. ALT+ Liposarcoma and the Mutation Status of ATRX/DAXX

The ALT+ phenotype is associated with LPS that have complex karyotypes involving chromosomal instability and multiple chromosomal gains and losses [177,179], whereas simpler karyotypes are associated with the TEL+ phenotype, including chromosomal translocations, for example t(12;16) [174,179]. Aggressive LPS subtypes, such as pleomorphic LPS and de-differentiated LPS, have been associated with mutations in the *ATRX* and *DAXX* genes [68,69,180]. Overall, the prevalence of ATRX alterations in LPS are about 20% and increase with tumor de-differentiation (*n* = 203) [68,69,180]. Specifically, ATRX was deficient in 24% (28/115) de-differentiated LPS [68,69,180], 41% (11/27) pleomorphic LPS [68,69,180], 0% (0/6) well-differentiated LPS [69], and 0% (0/55) myxoid LPS [68,69,180]. About 1% (*n* = 52) of de-differentiated LPS appear to be DAXX deficient [180]. On the other hand, ALT+ LPS experiences ATRX loss at a frequency of 78% [180], but if ATRX or DAXX are lost, LPS is ALT+ 100% of the time [68,180]. Loss of ATRX is more prevalent in LPS with complex karyotypes, which are known to have a higher frequency of ALT positivity [68,69].

#### 6.3.2. Liposarcoma and TMM-Guided Therapies

The past decade has brought several promising developments in LPS chemotherapy, including MDM2 antagonists, CDK4/6 inhibitors, and transcription factor binders, such as trabectedin [175,176]. However, TMM-stratified analysis or TMM-directed treatments in LPS are still lacking. At present, there are only three treatment studies that investigated LPS with variables related to TMM status [181,182,183]. While there are still no known effective targeted ALT therapies, these studies suggest that the existing drugs may be more effective for treating the TEL+ LPS.

For example, CDK4/6 and MDM2 inhibitors in co-treatment of seven LPS cell lines were found to require the presence of WT ATRX to induce senescence [176]. As ALT+ LPS predominantly lacks WT ATRX, these findings suggest that CDK4/6 therapy will likely be ineffective in treating the ALT+ LPS while more effective in treating the TEL+ LPS. The presence of ATRX directly decreases the expression of MDM2, transitioning LPS cells from quiescence into senescence [182]. This is suspected to occur via gene modulation, as ATRX/DAXX modulates chromatin structures, which likely then alter the gene expression of MDM2 [182]. If these potentially TMM-specific corollaries are corroborated, TMM biomarker screening will become essential to optimizing efficacy and future development of CDK4/6 inhibitors.

### 6.4. Undifferentiated Pleomorphic Sarcoma

Previously referred to as malignant fibrocystic histiocytomas, the undifferentiated pleomorphic sarcomas (UPS) encompass neoplasms that can arise in soft tissue and the dermis [184]. No true cell origin is known for UPS, and the histogenesis of UPS remains poorly defined [185]. UPS has thus been used as a wastebasket diagnostic term for tumors progressing toward undifferentiation [186]. Only after excluding all other diagnoses can a tumor receive this classification. UPS makes up about 10% of all soft tissue sarcomas and most commonly affects males between the ages of 50 and 70 [186]. Because tumor progression to UPS is still not understood, it is important to note any phenotypic similarities that might aid the proper diagnosis of these tumors. The ALT positivity may be such a characteristic.

Several studies have demonstrated ALT positivity in UPS [38,40,68,69,187]. The prevalence of ALT in UPS is about 59%, ranging between 36% and 77% (Table 2). A 2005 study found that there was no significant difference in patient survival between ALT+ and ALT− UPS patients [38]. However, a 2009 study by Matsuo and colleagues found ALT positivity to be the only independent prognostic factor in patient survival when compared to telomerase activity and tumor size [187]. Patients with ALT+/ TEL− UPS had a significantly worse prognosis than all other cohorts. Those with ALT−/TEL− UPS had the greatest survival rate. Interestingly, 19% (8/43) of this study’s UPS samples had both telomerase activity and ALT positivity (ALT+/TEL+) [187]. These preliminary findings suggest that ALT positivity may be useful in determining the prognosis of patients with UPS.

Overall, ATRX loss in UPS is about 37% (Table 3) and has been associated with ALT positivity [68,69]. In their diverse sarcoma panel, Koelsche and colleagues found that the prevalence of complete nuclear ATRX loss in UPS is higher than that of all other sarcomas (38%; 20/52) [69]. Though the prevalence of ATRX loss in UPS has only been assessed in two studies so far, the data suggests that UPS tumors may indeed be among the most susceptible to ATRX loss (Table 3 and Appendix A). ALT+ UPS experiences ATRX loss at a frequency of 55% [68], but if ATRX is lost, UPS is ALT+ nearly 96% of the time [68,69]. Furthermore, ATRX expression was retained in all ALT− UPS [68]. Intriguingly, Liau and colleagues found that only 20% (3/15) of radiation-associated sarcomas, which can be classified as UPS, or angiosarcoma, or leiomyosarcoma, were ALT+ and none of these tumors showed ATRX loss [68]. This suggests that the pathogenesis of radiation-associated sarcomas may be different from the de novo UPS.

## 7. The Development of Novel Therapies

There is a significant ongoing effort to develop therapeutics that selectively target the ALT pathway. However, as we will discuss, inhibitors with this selectivity have remained elusive. In brief, most therapeutic development has been tested in OS, although liposarcoma and glioma cell lines are increasingly tested. To date, there have been very limited or no studies that assess ALT targets in breast carcinoma, NB, PanNET, angiosarcoma, or LMS. This is surprising considering the size of the ALT literature. At present, four strategies held promise in treating ALT+ tumors. 

The first strategy is the use of ATR inhibitors against ALT+ cell lines, which initially showed great promise as a potential targeted therapy for ALT cancers [61]. However, a few recent studies challenged the specificity of ATR inhibitors towards ALT cancers [183,188,189]. ATR is an important kinase that functions in a variety of DNA damage response (DDR) pathways, including the activation of replication stress checkpoint and HR [190,191]. Initially, using 14 cancer cell lines (11 ALT+, 3 ALT−), Flynn and colleagues reported that an ATR inhibitor, VE-821, decreased the ALT phenotypes, including C-circles, APBs, and tSCEs, and selectively killed the ALT+ cells [61]. However, Deeg and colleagues later found opposing results when recapitulating the key experiments of this prior study [188]. Using the identical inhibitor (VE-821) as well as the same OS cell lines and assays, they argued that Flynn and colleagues’ original findings were confounded by a lack of control for the well-established variable growth rates of the assessed cell lines. Most recently, Goncalves and colleagues demonstrated that ATR inhibitor sensitivity in OS appears to be another TMM-independent therapeutic modality [189]. Using a panel of eight ALT+ OS and nine ALT− OS, they assessed the efficacy of three ATR inhibitors, AZD-6738, VE-822, and BAY-1895344. They made two key findings. First, the ALT− OS were more sensitive to the most potent ATR inhibitor, BAY-1895344. For other ATR inhibitors, there was no TMM-dependent statistical difference. Second, ALT− OS with short telomeres were the most sensitive to ATR inhibition [189]. These results are consistent with recent studies in Ewing Sarcoma, a ubiquitously ALT− bone tumor (Table 2), showing sensitivity to ATR inhibition [192] and similar ALT-independent ATR inhibitor sensitivity found in multiple soft tissue sarcomas including leiomyosarcoma, myxofibrosarcoma, and well-differentiated or de-differentiated LPS [183]. Using ATR inhibitor, VE-822, on eight sarcoma cell lines (four ALT+ and four ALT−), Laroche-Clary and colleagues found no significant difference in the IC_50_ values related to TMM status [183]. ATR-inhibition may not demonstrate TMM-specific efficacy in soft tissue tumors; however, co-treatment of ATR inhibitors with other cytotoxic agents, such as gemcitabine, has proven to be broadly effective by increasing DNA damage in a TMM-independent manner [183]. In contrast, ATR inhibition in breast cancer has been shown to be more cytotoxic to ALT+ cells [76]. In summary, tumor TMM is less likely to play a role in the effectiveness of ATR inhibition. It is still a hopeful treatment for patients with ALT+ tumors, though nonspecifically.

The second strategy is the application of G-quadruplex (G4) ligands. G4s are relatively stable nucleic acid secondary structures that are highly enriched in telomeres [58]. G4 ligands are organic molecules that can further stabilize the G4s. G4 ligands were initially shown to inhibit telomerase activity in TEL+ cancers, decrease cell proliferation and induce apoptosis [193,194]. However, studies in OS have found G4 ligands to be comparably effective at decreasing the viability of ALT+ cells (U2OS and Saos-2) and TEL+ cells (HOS) [195,196,197]. It is possible that in TEL+ and ALT+ cell lines, G4 ligands induce cell death by impeding telomerase activity and/or telomere replication.

The third strategy employs DNA topoisomerase 2 (TOP2) inhibitors. A recent study using yeast and U2OS cells suggests that selective TOP2 inhibition, with genistein, may kill ALT+ cells and not the ALT− ones [198]. TOP2 is thought to be required for telomere-telomere recombination and chromatin maintenance, phenomena both integral to the ALT mechanism [199,200]. Thus, TOP2 inhibition may cause excess telomeric replication stress, resulting in apoptosis. In vivo and in vitro knockdown of TOP2A and TOP2B as well as TOP2 inhibitor, ICRF-193, treatment reduced APB formation, increased TIFs, shortened telomeres, and hindered proliferation of ALT+ OS cells and ALT+ lung fibroblasts [200]. Similarly, two recent ellipticine derivatives, Z1 and Z2, were found to decrease C-circle and APB formation via TOP2 inhibition [201]. However, unlike genistein and ICRF-193, their anti-proliferative effects were also observed in TEL+ HeLa cells. Further complicating the matter, an in vitro study found doxorubicin, a known TOP2 inhibitor, was more effective against TEL+ pleomorphic LPS cells (SW872) than ALT+ pleomorphic LPS cells (LS2 and LiSa-2) [181]. The expression of TOP2A was measured in different LPS lines [181]. Interestingly, results indicated that the TEL+ LPS cells were more sensitive to doxorubicin than the ALT+ LPS cells, despite having less TOP2A. The resistance of ALT+ LPS to doxorubicin could be due to the downregulation of other topoisomerases or more active DNA repair. Overall, the current evidence surrounding the TMM-specific efficacy of TOP2 inhibitors remains conflicted.

A recent in vitro and in vivo study argues combination targeted ALT therapies would maximize therapeutic efficacy [202]. Such an approach to ALT tumors, which are predominantly high grade and likely to become treatment-resistant, would maximize potential clinical efficacy. To this end, George and colleagues found ATRX-aberrant NB cells are preferentially sensitive to irinotecan, a DNA topoisomerase 1 (TOP1) inhibitor, and Olaparib, a poly ADP ribose polymerase (PARP) inhibitor, respectively [202]. In combination, the dual therapy enhanced sensitivity but had a greater effect in ATRX-mutant cells. Furthermore, in a patient-derived xenograft model, one cycle of combination therapy was sufficient to induce sustainable remission and improved overall survival in ATRX-null mice [202]. Olaparib and irinotecan are already approved for use in clinical medicine, so there is great potential for this combination regiment to be studied in a patient setting.

The fourth strategy is oncolytic virotherapy. Sasaki and colleagues infected OS cells with an adenovirus, which relied on an hTERT promotor for replication, and found that the virus selectively killed OS cells in vivo and in vitro regardless of the TMM status [203]. Despite having low to absent expression of hTERT mRNA, ALT+ OS cell lines (U2OS, Saos-2, and HuO9) demonstrated ID_50_ values comparable to those of the nine TEL+ lines. In 10 out of the 12 cell lines tested, they found that the adenovirus vector induced a 1.1- to 50-fold increase in hTERT mRNA expression. Their data suggests that the efficacy of this potential treatment modality in ALT+ OS cells is driven by TMM-independent viral activation of the hTERT promoter. Recent findings by Han and colleagues suggest that Herpes Simplex Virus 1 (HSV-1) may have greater utility because HSV-1 infectivity relies on Infected Cell Polypeptide 0 (ICP0), a viral protein that disrupts PMLs [204]. Treatment of co-cultured ATRX-WT and ATRX-KO (ALT+) cells with WT HSV-1 resulted in a large and equivalent loss of both cell populations. Thus, implying WT HSV-1 has TMM-independent toxicity. However, interestingly treatment of the co-culture with ICP0-null HSV-1 resulted in an insignificant loss of ATRX-WT cells and a 61% loss of ATRX-KO cells [204]. Therefore, ICP0-null HSV1 holds great promise and merits further exploration as an ALT-specific therapy. Viral therapies rely on interacting with host receptors for efficacy, which cancers can manipulate to develop resistance. Therefore, continued design of oncolytic virotherapies with multiple receptor targets may be required. Despite this limitation, oncolytic virotherapy has established in vitro and in vivo efficacy, plus proven safety in phase I clinical trials, which may merit further investigation [203,205,206].

## 8. Conclusions

In recent years, many insightful discoveries have been made in elucidating the molecular mechanism of the ALT pathway. At the same time, a few novel and exciting molecular targets, including ATR, FANCM, and TOP2, have also been identified [25,27,188,200,201]. Recent small molecule screening suggests ATM inhibitor, KU60019; tyrosine kinase inhibitor, sunitinib; and HSP-90 inhibitor, 17-AAG are also promising [202]. Developing efficacious and targeted therapy for ALT cancers will naturally be the next step. However, mainly due to the discrepancies of different biomarkers used in identifying the ALT cancers, there is often a large range for the prevalence of ALT positivity in many cancers. To address this issue, we first reviewed and evaluated the commonly accepted and reliable biomarkers that can be used to identify the ALT positivity in both cell lines and tumor biopsies. Using a more stringent criterion, we then re-assessed the prevalence of ALT positivity in a variety of cancer types and, in most cases, were able to narrow down the range. Finally, based on this new survey of ALT cancers, we highlighted the important clinical ALT positive associations within each cancer type and subtype, their pathogenetic features, as well as their prognosis.

Most of the ALT+ cancers reviewed here appear to be of mesenchymal or neuroendocrine origin. ALT positivity has been very infrequently detected in tumors of epithelial origins. The only ALT+ cancer type reviewed here that is of potential epithelial origin is HER2+ breast carcinoma. However, the tissue hierarchy of the breast is very complex and contains many different epithelial cells as well as non-epithelial cells [207]. Therefore, the precise cell origin of ALT+ and HER2+ breast carcinoma warrants further clarification.

Collectively, when categorized based on different TMMs, ALT−/TEL− cancers seem to have the best prognosis, as illustrated in osteosarcoma, neuroblastoma, and undifferentiated pleomorphic sarcoma. The prognosis for the other three, ALT+/TEL+, ALT+/TEL−, and ALT−/TEL+, varies depending on the types/subtypes of cancers (Figure 1). For example, ALT positivity correlates with a worse prognosis for the HER2+ breast cancers, PanNETs, angiosarcoma, and all the soft tissue sarcomas, but with a better prognosis for osteosarcoma, astrocytoma, and neuroblastoma.

Pathogenetically, aberrations in the ATRX-DAXX-H3.3 pathway correlate highly with the ALT positivity (Table 3). Among the three genes frequently examined in this pathway, the large majority of the aberrations are due to that of the *ATRX* gene. When the loss of ATRX and DAXX is detected, the chance of the tumor being ALT positive varies from 83% in leiomyosarcoma to 100% in osteosarcoma and liposarcoma, suggesting that mutation analysis of the *ATRX* and *DAXX* gene is a reliable predictor of the ALT positivity in certain cancers. At present, this would be very helpful in predicting the prognosis of cancer patients. In light of the identification of a growing list of novel molecular targets, the targeted therapy for ALT cancers seems to be within reach. When that happens, together with the reliable ALT biomarkers, such as C-circle and APBs, the mutation status of *ATRX* and *DAXX* genes will also be valuable in predicting the drug response.

We hope that with this updated re-evaluation and clinical perspective, scientists, clinicians, and drug developers will gain greater insight into the ALT cancers so that together, we may develop novel and more efficacious treatments and better management strategies to meet the urgent needs of cancer patients.

## Figures and Tables

**Figure 1 cancers-13-02384-f001:**
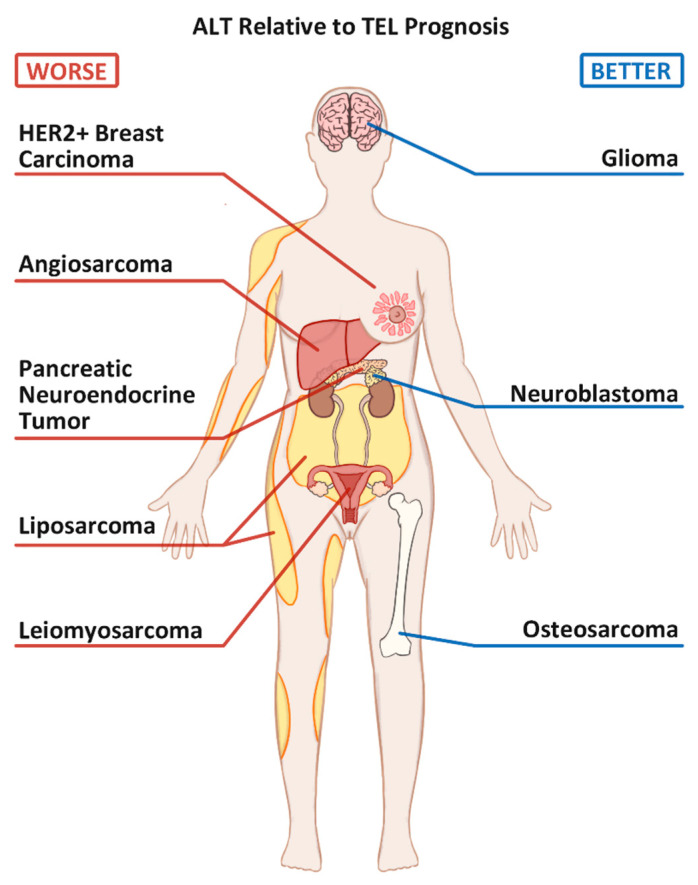
Commonly seen Alternative Lengthening of Telomeres positive (ALT+) cancers and their prognoses relative to their Telomerase positive (TEL+) counterparts.

**Table 1 cancers-13-02384-t001:** Markers used to distinguish cancers based on their telomere maintenance mechanism.

Biomarkers	Telo-FISH Nuclear Foci	APBs	TIFs	Telomerase Activity	C-circles	tSCE	Telomere Heterogeneity
ALT+	√	√	√	X	√	√	√
TEL+	X	X	X	√	X	X	X
Ever-Shorter Telomeres	X	X	X	X	X	X	X

ALT+: alternative lengthening of telomere (ALT) positive. TEL+: telomerase (TEL) positive. X: negative or relatively low; √: positive or relatively high; No shading: tissue sections; Light shading: tissue homogenate; Dark shading: tissue culture. Telo-FISH: Telomere (Telo) fluorescent in-situ hybridization (FISH). APBs: ALT-associated acute promyelocytic leukemia bodies. TIFs: telomere dysfunction-induced foci. tSCE: telomere sister chromatid exchange.

**Table 2 cancers-13-02384-t002:** Prevalence of ALT positivity in Human Cancers.

Tumor	% ALT+	Range *	Total Tumors Tested, *n*
**Bone**			
Chondrosarcoma	48%	N/A	31
Ewing Sarcoma	0%	N/A	62
Osteosarcoma	63%	49–86%	287
**Breast**			
Overall	4%	2–8%	552
HER2+ Breast Carcinoma	18%	14–21%	50
**Central Nervous System**			
Overall	20%	10–26%	4386
Glioma	30%	13–69%	912
NF1 loss-associated glioma	44%	29–69%	167
Astrocytoma (Overall)	23%	10–78%	2231
Diffuse astrocytoma (grade II)	55%	27–100%	95
Anaplastic astrocytoma (grade III)	65%	21–92%	118
Adult glioblastoma (grade IV)	16%	11–25%	862
Pediatric glioblastoma (grade IV)	39%	12–56%	89
Oligodendroglioma	17%	0–25%	78
Oligoastrocytoma	60%	11–72%	48
Ependymoma	0%	N/A	260
Medulloblastoma	7%	2–22%	237
Choroid plexus carcinomas	23%	N/A	31
**Neuroendocrine**			
Neuroblastoma	24%	18–47%	843
Pancreatic Neuroendocrine Tumor (PanNET)	32%	21–61%	1152
**Soft Tissue**			
Angiosarcoma	23%	11–24%	79
Leiomyoma	3%	0–3%	217
Leiomyosarcoma	60%	52–78%	331
Liposarcoma	27%	25–29%	566
Undifferentiated Pleomorphic Sarcoma	59%	36–77%	174

*: Minimum and maximum of horizontal range is per the mean ALT+ prevalence acquired of CCA, APB, Telo-FISH (alone), and TRF, respectively (see Appendix A). N/A: not applicable – horizontal range could not be generated as this tumor was only assessed via one biomarker.

**Table 3 cancers-13-02384-t003:** ATRX/ DAXX loss in ALT-associated human cancers.

Tumor	ATRX Loss Only, *n*	DAXX Loss Only, *n*	Both ATRX and DAXX Loss, *n*	Total Tumors Tested, *n*	Total Mutant % *	ATRX/DAXX Mutant Cases that are ALT+, %	ALT+ Cases that are ATRX Mutant, %
**Bone**							
Chondrosarcoma	0 **	N/A ***	N/A	15	0%	N/A	N/A
Ewing Sarcoma	0	N/A	N/A	12	0%	N/A	N/A
Osteosarcoma	17	0	0	71	24%	100%	58%
Breast							
Breast Carcinoma	0	0	0	96	0%	N/A	N/A
**CNS**							
Glioma	403	7	N/A	1607	26%	74%	71%
**Neuroendocrine**							
Neuroblastoma (NB)	83	1	0	1052	8%	92%	67%
High Risk NB	25	0	0	165	15%	N/A	100%
PanNET	218	153	23	1223	32%	96%	86%
**Soft Tissue**							
Angiosarcoma	16	0	2	77	21%	N/A	88%
Leiomyoma	6	1	0	206	3%	43%	67%
Leiomyosarcoma	103	4	0	311	34%	83%	56%
Liposarcoma (LPS)	39	1	N/A	203	20%	100%	78%
Well differentiated LPS	0	N/A	N/A	6	0%	N/A	N/A
Dedifferentiated LPS	28	1	0	52	56%	100%	93%
Myxoid LPS	0	N/A	N/A	55	0%	N/A	N/A
Pleomorphic LPS	11	N/A	N/A	27	41%	100%	63%
Undifferentiated Pleomorphic Sarcoma	32	N/A	N/A	87	37%	96%	55%

*: Total Mutant % = (ATRX loss + DAXX loss + Both loss)/Total Tumors; **: 0 indicates that at least one study assessed for the gene, but gene loss was not observed in any samples; ***: N/A indicates that the data was not available or the study did not assess for the gene of interest. ATRX: α-thalassemia/mental retardation syndrome X-linked (*ATRX*) gene. DAXX: death-domain associated protein gene.

## Data Availability

The data presented in this study are available in the Appendix A.

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
