# Peer review of "ALT Positivity in Human Cancers: Prevalence and Clinical Insights"

_cancers, 2021, doi:10.3390/cancers13102384_

Round 1

Reviewer 1 Report

In this manuscript MacKenzie Jr. and colleagues perform a comprehensive review of ALT positivity in human cancers. Although several recent reviews have been written on ALT cancers the focus of the authors on the prevalence and clinical insights is certainly of value and will be of great utility and interest for the field. The review is structured in distinct sections and addresses each disparate cancer type in turn. The structure is good, however, lacks some consistency as the authors explore therapeutic approaches for some of the cancer types but not others. To address this the authors may wish to consider including a separate section on the therapeutic targeting of ALT cancers due to the overlap between cancer types, this, however, is largely an editorial decision. The review is largely well written, with some minor corrections required prior to publication. Some important references should also be included that have not been covered. Figure 1 and Extended Table 2 appear to be missing from this submission; hence I have not been able to assess them. More detailed comments are given below:

Line 16 – the authors state ‘the greatest flurries have been with respect to the alternative lengthening of telomeres (ALT) pathway’. Flurries is an orthodox term to use here – perhaps ‘advances’ would be more appropriate.

Line 17 – Assessing should be replaced with assess.

Line 41 – preventing should be replaced with prevent.

The authors may wish to add a sentence on the end replication problem to explain to the reader why telomeres in somatic cells progressively shorten.

Line 89 – The authors state ‘To best review the recent surge in ALT prevalence literature, careful evaluation of how ALT is identified becomes quite necessary.’ I would argue this should read very important as opposed to quite necessary.

On line 168 the authors discuss that C-circles are likely the by-products of BITS during the repair and re-start of the stalled or collapsed replication forks at telomeres. The authors should discuss the work of Mazzucco and co-workers here who have identified I-loops as a likely precursor for extrachromosomal telomeric circles (PMID: 33082350)

The authors discuss commonly used biomarkers for identifying ALT but do not discuss the visualisation of telomeric DNA synthesis in G2 or mitosis which is perhaps the most direct assessment of ALT positivity via the ATSA assay (PMID: 30673617).

On line 202 the authors introduce the use of telomeric variant repeats as a predictor for TMM. The authors should state what sequences exactly were being looked at exactly here.

On line 258 the authors state that ‘ATRX interacts with DAXX and together they function to remodel chromatin and deposit certain histone variants, like H3.3, onto heterochromatin’. What other histone variants was the author referring to here, the only other variant ATRX has been reported to influence the deposition of is macroH2A (PMID: 30833786).

On line 301 it is not clear why this is a parsimonious study?

In the discussion regarding the use of oncolytic viruses to target Osteosarcoma the authors should discuss the work of Reddel and colleagues in using HSV-1 lacking ICP0 to target ATRX deficient cancer cells (PMID: 30745338).

In section 3 the authors discuss that poor prognosis of HER2+ breast cancers may be associated with an increased expression of the SLX4 interacting protein, SLX4IP. The authors should comment here that this finding is somewhat in contrast to work reported by Panier and colleagues, which suggested that SLX4IP is inactivated in a subset of ALT positive Osteosarcomas and its loss is associated with an increase in ALT-related phenotypes (PMID: 31447390).

On line 484 the authors state that loss of XRCC1 leads to impaired NHEJ. The authors need to clarify here that XRCC1 loss leads to an impairment of alternative NHEJ and not the canonical NHEJ pathway.

When discussing genetic alterations and biomarkers associated with ALT positive gliomas the authors should comment on SETD2 mutations which have been identified as mutually exclusive with H3.3 G34R/V mutations (PMID: 23417712).

When discussing possible treatments for ALT positive gliomas the authors should discuss the work of Bo Han and colleagues which showed an enhanced sensitivity to temozolomide upon ATRX, owing to suppression of ATM mediated DNA damage repair (PMID: 29378238).

In section 5 the authors do not include a section on the potential treatment strategies for ALT positive neuroblastomas. Recent work by George and co-workers has highlighted selective sensitivity in ATRX mutant neuroblastoma cells to multiple PARP inhibitors the ATM inhibitor KU60019 and various Topoisomerase I inhibitors. This should really be commented upon (PMID: 32846370).

Reviewer 2 Report

I have read with interest the very detailed review by MacKenzie et al about the ALT mechanism of telomere maintenance: its detection, its distribution among cancer types, the associated genetic alterations and the perspectives for future novel anti-ALT therapies. Although the reviewing exercise is remarkable, I felt that the manuscript was too long, with redundant paragraphs. I am mostly referring to the description of ATRX/DAXX alterations that appears at various places of the manuscript. My opinion would be that one dedicated chapter to these alterations –that are commonly found in ALT+ cancers- would be better than repeating this for each cancer type. I also found that the chapter dedicated to the development of potential novel therapies, that, in this version of the manuscript, appears under the osteosarcoma section, would better fit in a dedicated separate chapter at the end of the manuscript as, again, it is likely that anti-ALT therapies may share lots of features across cancer types. That ATR inhibitors for instance were tested in OS cell lines does not restrict them to this cancer type. So, globally, I would prefer to have a dedicated chapter at the end for all cancer types.

I am detailing below more specific concerns:

  • Page 2. Stating that, in TEL+ cells, telomere length follows a Gaussian distribution with a mean of 10-15 kb is absolutely not true as, unlike normal cells, TEL+ cancer cells show very distinct telomere lengths, with telomeres that can be very short sometimes. This should be changed.
  • The tools used for the detection of ALT+ phenotype in tumors have been recently reviewed by Claude and Decottignies (2020) and the paper may be cited.
  • I did not have Fig 1 and Extended Table 2 attached to the manuscript when downloading for review and I therefore cannot evaluate them.
  • Table 1: “Telo-FISH” as marker does not mean much. This should be changed. Telomere heterogeneity: assessed by Telo-FISH? “Ever shortening of telomeres” appears in the table without any associated reference. If you refer to the back-to-back studies published in Cell Reports in 2017 (Dagg et al, Viceconte et al), this should appear. I also find that this table does not give a good idea of the markers that can be used on tumor sections and those that rely on cancer cell cultures. The ever shortening of telomeres phenotype for instance can only be evaluated with cultured cells. Same for tSCE. So, overall, I feel that Table 1 should be revisited.
  • Throughout the manuscript, you should have uniform nomenclatures for ALT+ cells or tumors. Sometimes, they are called “ALT positive”, or simply “ALT”. Please check this carefully.
  • On many instances, the text details too much the numbers that can be easily appreciated in the tables. For instance, listing, in the text, the ranges of ALT+ prevalence for each tumor type is not necessary as readers have the numbers in Table 2. Overall, the text can be shortened.
